# The Associations between Meeting 24-Hour Movement Guidelines (24-HMG) and Mental Health in Adolescents—Cross Sectional Evidence from China

**DOI:** 10.3390/ijerph20043167

**Published:** 2023-02-10

**Authors:** Lin Luo, Xiaojin Zeng, Yunxia Cao, Yulong Hu, Shaojing Wen, Kaiqi Tang, Lina Ding, Xiangfei Wang, Naiqing Song

**Affiliations:** 1College of Physical Education, Guizhou Normal University, Guiyang 550001, China; 2Basic Education Research Center, Southwest University, Chongqing 400715, China; 3Research Institute of Sports Science, Wuhan Sports University, Wuhan 430079, China

**Keywords:** 24-hour movement guidelines, physical activity time, screen time, depression, adolescents, anxiety

## Abstract

(1) Background: This study determined the prevalence of adolescents that meet 24-HMGs alone and in combination, and their association with the risk of developing adolescent anxiety and depression. (2) Methods: Participants were drawn from 9420 K8 grade adolescents (age 14.53 ± 0.69 years; 54.78% boys) from the China Education Tracking Survey (CEPS) 2014–2015 tracking data. Data on depression and anxiety were collected from the results of the questionnaire in the CEPS for the adolescent mental health test. Compliance with the 24-HMG was defined as: physical activity time (PA) ≥ 60 min/day was defined as meeting the PA. Screen time (ST) ≤ 120 min/day was defined as meeting the ST. Adolescents aged 13 years achieved 9–11 h of sleep per night and adolescents aged 14–17 years achieved 8–10 h of sleep per night, defined as meeting sleep. Logistic regression models were used to examine the association between meeting and not meeting the recommendations and the risk of depression and anxiety in adolescents. (3) Results: Of the sample studied, 0.71% of adolescents met all three recommendations, 13.54% met two recommendations and 57.05% met one recommendation. Meeting sleep, meeting PA+ sleep, meeting ST + sleep, and meeting PA + ST + sleep were associated with a significantly lower risk of anxiety and depression in adolescents. Logistic regression results showed that differences in the effects of gender on the odds ratio (ORs) for depression and anxiety in adolescents were not significant. (4) Conclusions: This study determined the risk of developing depression and anxiety in adolescents who met the recommendations for 24-HMG alone and in combination. Overall, meeting more of the recommendations in the 24-HMGs was associated with lower anxiety and depression risk outcomes in adolescents. For boys, reducing the risk of depression and anxiety can be prioritised by meeting PA + ST + sleep, meeting ST + sleep and meeting sleep in the 24-HMGs. For girls, reducing the risk of depression and anxiety may be preferred by meeting PA + ST + sleep or meeting PA+ sleep and meeting sleep in 24-HMGs. However, only a small proportion of adolescents met all recommendations, highlighting the need to promote and support adherence to these behaviours.

## 1. Introduction

Previous studies have extensively investigated the relationship between adequate physical activity, limited screen time and adequate sleep time, as well as the physical and mental health of adolescents [1,2,3,4,5]. More recently, physical activity time, screen time and sleep time have been proposed jointly in relevant studies as three key components of the 24-hour movement guidelines because of their interactions and interdependence [6,7]. Using the subjects’ use of time of day as an entry point for the study [8], the researchers proposed a framework for the study. This framework integrates physical activity time, screen time and sleep time, as this combination provides more reliable information on health and behaviour changes [9]. In Australia and Canada, 24-hour movement guidelines (24-HMG) for children and adolescents containing three recommendations for physical activity time, screen time and sleep time were published in 2015 and 2016, respectively [6,7]. Both versions of the 24-HMG recommend that children and adolescents engage in at least 60 min of moderate-to-vigorous physical activity per day. Screen time should not exceed 2 h per day. Children and adolescents aged 5–13 years should have at least 9–11 h of sleep per night, and adolescents aged 14–17 years should have at least 8–10 h of sleep per night [10]. A growing amount of research evidence showed that adolescents who consistently meets 24-HMG can achieve additional health benefits, such as preventing obesity, improving emotional health, improving metabolic indicators and promoting social adjustment [11,12,13].

Depression and anxiety are common mental health problems among adolescents, and depression is often accompanied by anxiety. From 2001 to 2020, the global prevalence of depression among adolescents increased by 34%, with the highest prevalence of depression in the Middle East, Africa and Asia [14]. In a survey of 38 countries/regions, Erskine et al. reported a 6.2% prevalence of depression and a 3.2% prevalence of anxiety in children and adolescents aged 5–17 years [15]. Previous studies have shown that the three recommendations of adequate physical activity (PA ≥ 60 min/day) [16,17], limited screen time (ST ≤ 120 min/day) [18,19] and good sleep duration (meeting age-specific needs) [20,21] in the 24-HMG are associated with lower depression and anxiety symptoms in adolescents, respectively. For example, a study by McDowell et al. (2017) reported that adolescents aged 14–17 years (*n* = 481) who achieved PA ≥ 60 min/day had 30% lower odds of depression and 46% lower odds of anxiety compared to adolescents with PA < 60 min/day (insufficient time or insufficient days) [22]. A meta-analysis study conducted by Mougharbel et al. (2020) showed that higher screen time was associated with more severe depressive symptoms in adolescents who engaged in ST > 120 min/day. However, its relationship with anxiety symptoms was not stable and may be related to gender [23]. A study by Baum et al. (2014) found that insufficient sleep duration affected anxiety and fatigue in healthy adolescents aged 14–17 years [24]. Ojio et al. (2020) showed that weekday sleep duration was independently associated with levels of depressive symptoms in Japanese adolescents aged 12–15 years (*n* = 942) [25]. Thus, adolescents who have adequate physical activity time, less screen time and adequate sleep time may have better mental health and be less likely to experience anxiety and depressive symptoms [12]. However, these behavioural studies mainly examined the recommendations in isolation and not the combined effects on adolescent depression and anxiety.

With the release of the 24-HMG, more researchers have begun to examine adolescent mental health from the perspective of integrating physical activity time, screen time and sleep time, rather than examining multiple determinants in isolation. For example, a systematic evaluation by Sampasa-Kanyinga et al. (2020) of 115,540 adolescents aged 5–17 years in 12 countries, including the USA and the UK, found that meeting the three recommendations in the 24-HMG held a good association with better mental health indicators (depression and other mental health indicators) [26]. The study by Lu et al. (2021) of 5357 adolescents in grades K4 and K5 in China found that meeting the 24-HMG recommendations was associated with better mental health outcomes [27]. It has been suggested that holistic surveys that integrate physical activity, screen time and sleep time may improve their validity in predicting individual health outcomes [8]. From a practical application perspective, it may be possible to compare the differences in meeting the impact of different recommendations on adolescent mental health [28]. Despite the benefits of meeting the 24-HMG on adolescent mental health, research evidence is still very limited, which may reduce the generalizability of the findings. Therefore, there is a need to replicate and extend this study in other culturally diverse populations and across age groups to further confirm the association between 24-HMG and adolescent mental health. While previous studies have conducted preliminary explorations of these factors as key to adolescent mental health in small samples [26], the use of a nationally representative sample in this study will help to improve the generalisability of the findings, which can inform policy makers when designing effective interventions. Several previous studies have found that there may be gender differences in the association of these factors with mental health. Therefore, the aims of this study included: (a) to examine the relationship between the 24-HMG recommendations met by Chinese adolescents and the risk of depression and anxiety disorders; and (b) to determine whether this association was influenced by gender.

## 2. Materials and Methods

### 2.1. Study Design and Participants

This study analysed primary data from the China Education Tracking Survey (CEPS), a national cohort study designed to document the attributes of developmental and educational experiences of Chinese adolescents. The CEPS study adopted a multi-stage stratified design with county (or equivalent administrative area), school and class as primary, secondary and tertiary sampling units, respectively. Primary sampling units were stratified by region and size of the migrant population, with counties in Shanghai or other regions with high migrant populations being oversampled. Within each sampled county, four schools were sampled using the probability proportion method. the CEPS team developed survey weights to address the probability of inequality of choice [29]. This study used the CEPS data from 2014–2015. The study involved data on youth and parent demographic characteristics, physical exercise time, screen time, sleep time, and depression and anxiety scores. The 2014–2015 CEPS youth (grade K8) sample size was 10,750. A total of 301 classes were surveyed in the 2014–2015 survey. This included a sample of 9653 for physical activity time, 9865 for screen time, 9649 for sleep time and a sample size of 9888 for the mental health data. Excluding data with missing physical exercise time, screen time, sleep time, depression and anxiety scale scores, the final study sample size for this study was 9420. This study used the free public database of the CEPS. The ethical review of the CEPS was approved by the ethics committee of the People’s University of China. Prior to the CEPS, all participants signed a consent form and their parents also signed a consent form to participate in the study. Students were not rewarded for their participation. Details of the CEPS data, and information on its ethical review, can be found at http://ceps.ruc.edu.cn/, accessed on 19 January 2023.

### 2.2. Measures

This study used the Canadian 24-HMG as a reference standard. The contents of the Canadian 24-HMG are shown in Table 1. This study used time for moderate to vigorous physical exercise from the physical activity recommendations, sleep time from the sleep recommendations and screen time from the sedentary recommendations [30]. Then, we analysed their relationship with depression and anxiety in adolescents.

#### 2.2.1. Physical Activity Time

The CEPS investigated the frequency of physical activity per week and the duration of a single session of physical activity in adolescents, and we referenced the Canadian 24-HMG, where PA ≥ 60 min/day was defined as meeting physical activity guidelines (meeting PA).

#### 2.2.2. Screen Time

The CEPS investigated the daily screen time of adolescents from Monday to the weekend. The amount of time adolescents spent watching television, surfing the internet and playing computer games each day was aggregated to obtain the amount of screen time adolescents spent each day. Referring to the Canadian 24-HMG, screen time ≤ 120 min/day was defined as meeting sedentary guidelines (meeting ST).

#### 2.2.3. Sleep Time

The CEPS surveyed adolescents’ self-reported daily sleep duration. Referring to the Canadian 24-HMG, satisfying sleep was defined as sleeping at least 9–11 h per night for 13 year olds and 8–10 h per night for 14–17 year olds.

#### 2.2.4. Depression and Anxiety Scores

The CEPS measures anxiety and depression in adolescents with a total of nine items. The scale was designed with reference to the generalized anxiety scale (GAD-7) [31,32] and the health questionnaire depressive symptom scale (PHQ-9) [33]. Three of the questions measured anxiety, and Cronbach’s alpha coefficient for this questionnaire was 0.815. Six questions measured depression, and Cronbach’s alpha coefficient for this questionnaire was 0.918. Raw scores for the anxiety-related questions were summed to obtain a total score for anxiety. The raw scores for the depression-related questions were summed to obtain a total score for depression. Referring to the previous literature, we defined the score at position P_75_ as the critical score. A total anxiety score exceeding the sample P_75_ score was defined as having anxiety symptoms. A total depression score exceeding the sampleP_75_ score was defined as having depressive symptoms [34].

#### 2.2.5. Covariates

Study participants were asked to self-report demographic data, including sex, age, ethnicity, single child, residence, father’s highest education, mother’s highest education, body mass index (BMI) and perceived household economic status. BMI was calculated by dividing weight (kg) by the square of height (m) [35]. In this study, WS/T 586-2018 screening criteria for overweight and obesity in school-aged children and adolescents were used. This is a set of Chinese national standards (WS/T 586-2018) recommended by the National Health Council of China for adolescents aged 6 to 18 years [36]. BMI is classified as not overweight (including low and normal weight) and overweight (including overweight and obesity).

### 2.3. Statistical Analysis

All data analyses were carried out using Stata 17.0 software. Descriptive statistics were used to characterise the sample. Frequency percentages were used to describe the categorical variables. Continuous variables were tested with the Shapiro–Wilk test, satisfying the normality distribution, and the variables were described by the mean ± SD. Gender differences between variables were performed using chi-square tests or t-tests. Logistic regression analysis was used to test the relationship between physical activity time (PA ≥ 60 min/day), screen time (ST ≤ 120 min/day) and sleep time (9–11 h at age 13, 14–17 years—8–10 h) in meeting the 24-HMG and the occurrence of anxiety and depression. Adolescents were divided into eight groups based on their meeting of the 24-HMG recommendations: meeting none, only meeting PA, only meeting ST, only meeting sleep, meeting PA + ST, meeting PA + sleep, meeting ST + sleep, meeting PA + ST + sleep. All logistic regression models were adjusted for covariates and analysed extraordinarily to see if this association was affected by gender. Statistical significance was defined as *p* < 0.05.

## 3. Results

### 3.1. Sample Characteristics

The final study sample size for this study was 9420 Chinese adolescents in grade K8. The mean age was 14.53 ± 0.69 years. The overall sample consisted of 54.81% boys and 44.86% single children (Table 2). Participants had a 29.44% prevalence of anxiety and a 26.93% prevalence of depression. The proportion meeting the recommendations for physical activity time, screen time and sleep time were 5.31%, 20.32% and 60.54%, respectively. Only 0.71% of participants met all three recommendations in the 24-HMG, 13.54% met any two recommendations, 57.05% met one guideline and 28.70% did not meet any of the recommendations. A higher proportion of boys than girls met all three recommendations in the 24-HMG.The differences between boys and girls were significant in terms of age, ethnicity, single child, residence, father’s highest education, mother’s highest education, perceived household economic status, BMI, meeting PA, meeting ST, meeting sleep and meeting 24-HMG categories. There were no significant differences in the prevalence of anxiety and depression. A description of participant characteristics by gender is shown in Table 2.

### 3.2. Prevalence of Depression and Anxiety among Adolescents in Different 24-HMG Categories

The prevalence of depression and anxiety in different groups of adolescents is shown in Figure 1 and Figure 2. Overall, the three groups with the lowest incidence of depression were, in order, the meeting PA + ST + sleep, meeting PA + sleep group, and meeting ST + sleep group. The three groups with the lowest incidence of anxiety were, in order, the meeting PA + ST + sleep, meeting PA + ST group, and meeting ST + sleep group.

Among boys, the three groups with the lowest prevalence of depression were, in order, the meeting PA + ST + sleep group (9.43%), the meeting ST + sleep group (21.15%) and the meeting PA + sleep group (22.68%). Among the girls, the three groups with the lowest prevalence of depression, in order, were the meeting PA + ST group (11.11%), the meeting PA + sleep group (212.50%) and the meeting ST + sleep group (19.56%).

Among boys, the three groups with the lowest prevalence of anxiety were, in order, the meeting PA + ST + sleep group (7.55%), the meeting ST + sleep group (22.44%) and the meeting PA + sleep group (23.71%). Among the girls, the three groups with the lowest prevalence of anxiety were, in order, the meeting PA + sleep group (9.8%), the meeting ST + sleep group (21.22%) and the meeting PA + ST + sleep group (21.43%).

### 3.3. Meeting One of the 24-HMG Recommendations in Relation to Anxiety and Depression

Table 3 presents the relationship between meeting one of 24-HMG recommendations and adolescent anxiety and depression obtained through a binary logistic regression. The results of the study showed that the ORs for depression and anxiety in the only meeting sleep group were 0.65 (*p* < 0.01) and 0.68 (*p* < 0.01), respectively, compared to the not-meeting group. This indicated that the effect of only meeting sleep on depression was more significant than the effect on anxiety. Logistic regression results showed that differences in the effects of gender on the ORs for depression and anxiety in adolescents were not significant. In the boys sample, the ORs for depression and anxiety in the only meeting sleep group were 0.60 (*p* < 0.01) and 0.64 (*p* < 0.01), respectively. In the girls sample, the ORs for depression and anxiety in the only meeting sleep group were 0.70 (*p* < 0.01) and 0.73 (*p* < 0.01), respectively.

### 3.4. Meeting Two of the 24-HMG Recommendations in Relation to Anxiety and Depression

Table 4 presents the relationship between meeting two 24-HMG recommendations and adolescents’ anxiety and depression obtained through a binary logistic regression. The results of the study showed that the ORs for depression and anxiety in the meeting PA + sleep group were 0.52 and 0.59, respectively, compared to the meeting none group (*p* < 0.01). This indicates that the effect of meeting PA and sleep guidelines on depression was more significant than the effect on anxiety. Compared to the meeting none group, the ORs for depression and anxiety in the meeting ST + sleep group were 0.46 and 0.57, respectively (*p* < 0.01). Logistic regression results showed that differences in the effects of gender on the ORs for depression and anxiety in adolescents were not significant.

In the boys sample, the ORs for depression and anxiety in the meeting PA + sleep group were 0.55 and 0.64, respectively, compared to the meeting none group (*p* < 0.01). This indicates that meeting PA + sleep had a more significant effect on depression than on anxiety. The ORs for depression and anxiety in the meeting ST + sleep group were 0.42 and 0.54, respectively, compared to the meeting none group (*p* < 0.01). For boys, this indicates that the effect of meeting ST and sleep guidelines on depression was more significant than the effect on anxiety. The effect of meeting ST and sleep guidelines on depression and anxiety was more significant than the effect of meeting PA and sleep guidelines on depression and anxiety.

In the girls sample, the ORs for depression and anxiety in the meeting PA + sleep group were 0.29 and 0.24, respectively, compared to the meeting none group (*p* < 0.01). This indicates that the effect of meeting PA and sleep guidelines on anxiety was more significant than that on depression. Compared to the meeting none group, the ORs for depression and anxiety in the meeting ST + sleep group were 0.42 and 0.59, respectively (*p* < 0.01). For girls, This indicates that the effect of meeting ST and sleep guidelines on depression was more significant than the effect on anxiety. The effect of meeting PA and sleep guidelines on depression was more significant than the effect of meeting ST and sleep guidelines on depression. The effect of meeting ST and sleep guidelines on anxiety was more significant than the effect of meeting PA and sleep guidelines on anxiety. The effect of meeting PA + and sleep guidelines on depression was more significant than the effect of meeting PA + and sleep guidelines on anxiety. The effect of meeting ST and sleep guidelines on anxiety was more significant than the effect of meeting PA and sleep guidelines on anxiety.

### 3.5. Meeting Three of the 24-HMG Recommendations in Relation to Anxiety and Depression

Table 5 presents the relationship between meeting the three 24-HMG recommendations and adolescents’ anxiety and depression obtained through a binary logistic regression. The findings show that the ORs for depression and anxiety in the meeting PA + ST + sleep group were 0.27 and 0.18, respectively, compared to the meeting none group (*p* < 0.01). This indicates that meeting PA, ST and sleep guidelines had a more significant effect on anxiety than on depression. In the boys sample, the ORs for depression and anxiety in the meeting PA + ST + sleep group were 0.20 and 0.08, respectively, compared to the meeting none group (*p* < 0.01). This indicates that meeting PA, ST and sleep guidelines had a more significant effect on anxiety than on depression. In the girls sample, the ORs for depression and anxiety in the meeting PA + ST + sleep group were 0.52 and 0.60, respectively, compared to the meeting none group (*p* < 0.01). This indicates that the effects of meeting PA, ST and sleep guidelines were more significant for anxiety than for depression. Logistic regression results showed that the differences in the effects of gender on the ORs for depression and anxiety in adolescents were not significant.

## 4. Discussion

This study used a K8 grade sample based on the CEPS (2014–2015) to determine the relationship between meeting the 24-HMG and students’ self-rated anxiety and depression. In this study, only 0.71% of the adolescents in the study sample met the three recommendations of the 24-HMG. This indicated a low adherence to the guideline among Chinese adolescents in grade K8 compared to previous studies. A previous study by Liu et al. (2022) of 14–17 year old adolescents in grades K9-K12 in the USA found that the proportion of adolescents meeting all three of the 24-HMG was 3% [37]. The study of Janssen et al. (2017) of 17,000 Canadian adolescents aged 10–17 years found that less than 3% of adolescents met all three of the 24-HMG [38]. In a study of 3772 Spanish minors aged 4–14 years, López-Gil et al. (2022) found that the proportion of children and adolescents meeting all three of the 24-HMG was 13.5% [39]. This result was also lower than that reported in a previous study by Chen et al. (2021). Their study of 114,072 Chinese adolescents aged 6–13 years in grades K4-K12 found that the proportion of adolescents meeting all three of the 24-HMG was 5% [40]. The reason for the inconsistency between the present study and the results of these previous studies, however, may be related to the grade range of the population from which the sample was collected and the different testing instruments. For example, some studies used accelerometers to estimate the PA status [8], while others used self-reported PA data (as in this study). In this study, there was a gender difference in the proportion meeting the recommendations in 24-HMG. Previous studies have reported that among younger children, boys were more likely to meet all three recommendations than girls. However, among adolescents, there were no gender differences in meeting all three recommendations. However, regarding the components of the 24-HMG, adolescent boys were more likely than girls to meet the physical activity recommendations, and females were more likely than males to meet the screen time recommendations (Sampasa-Kanyinga et al., 2020) [41]. In this study, boys were more likely than girls to meet the PA and sleep recommendations, and girls were more likely than boys to meet the ST recommendations, but a higher proportion of boys met all three recommendations. Previously, Ying et al. (2020) did not find gender differences in meeting sleep and screen time, in a survey of Chinese high school students meeting the 24-HMG recommendations [42]. Therefore, the relationship between meeting the 24-HMG recommendations and gender needs to be further developed in future studies. However, it is worth noting that low adherence to 24-HMG among Chinese adolescents may threaten clinical and public health outcomes. Previous studies have found that compliance with 24-HMG has beneficial effects on adolescent health development. Therefore, this issue should be addressed through an effective public health approach. However, there is limited evidence on how to optimise adolescent health behaviours and therefore future research on intervention strategies to promote adolescent compliance with 24-HMG should be encouraged.

In a study by de Castro et al. (2023) on depression and anxiety among adolescents in 26 low- and middle-income countries, 5.5% of adolescents (*n* = 123,975, 10–17 years) had symptoms of anxiety and 3.1% had symptoms of depression [43].In a study by Merikangas et al. (2010), the prevalence of anxiety among adolescents in the United States (*n* = 10, 123, 13–18 years) was 31.9% [44]. Ma et al. (2021) reported a prevalence of depression and anxiety of 29% and 26%, respectively, in adolescents (*n* = 57,927) [45]. The differences in the prevalence of depression and anxiety among adolescents in these studies were related to the depression and anxiety testing instruments used in the different studies. In the present study, 26.93% and 29.44% of adolescents self-reported symptoms of depression and anxiety, respectively, and no significant gender differences in the prevalence of depression and anxiety were observed. This data, is close to the previous study by Tang et al. (2019). Their study of Chinese adolescents (*n* = 144,060) reported a 24.3% (95% CI, 21.3%–27.6%) prevalence of depression in adolescents [46].

Previous studies had systematically evaluated the independent effects of physical activity, screen time and sleep on adolescent mental health outcomes [16,17,18,19,20,21,22]. In contrast, not many studies evaluated their combined effects of the three jointly on adolescent mental health. Consistent with past studies [9,47,48], our study showed that meeting all three of the 24-HMG was associated with a lower risk of anxiety and depression in adolescents. These studies highlighted the importance of helping adolescents meet more of the 24-HMG recommendations, which could be considered as a way to prevent or treat mental health problems in adolescents in this age group. In line with the findings of Zhu et al. (2019), the present study confirmed that the higher the number meeting the 24-HMG recommendations, the lower the odds of anxiety and depression in adolescents [48]. This finding is indirectly supported by the study of Sampasa-Kanyinga et al. (2017) [9], showing that adolescents who met the optimal combination of PA, screen time and sleep time had the lowest odds of depression or anxiety. This is consistent with the findings of Lu et al. (2021) in a study of Chinese children aged 10–13 years [27]. The results of this study suggested that meeting the 24-HMG guidelines would be beneficial in maintaining better mental health in Chinese adolescents at the junior high school level. These findings may further help refine and update the cross-cultural use of the 24-HMG.

This study found that meeting sleep recommendations significantly reduced the risk of depression and anxiety in adolescents. The relationship between insufficient sleep duration and anxiety and depression has been confirmed in many studies [49,50]. Insomnia is the most common sleep disorders among adolescents (Johnson et al., 2006) [51]. Inadequate sleep duration in adolescents had serious implications for future health and functioning (Brand et al., 2009) [52] and was thought to trigger and maintain many emotional and behavioural problems, particularly anxiety and depression (Dahl and Harvey, 2007) [53]. Sleep, arousal and emotion represent overlapping regulatory systems, with dysregulation of one system affecting the others, so that sleep disruption during critical periods of maturational development may provide a pathway for later emotional dysregulation, and vice versa (Dahl, 1996) [54]. Sleep deprivation has been shown to increase negative emotions, decrease positive emotions and alter the way adolescents understand, express and regulate their emotions (Palmer et al., 2016) [55]. Several studies have shown an increase in depression (Fredriksen et al., 2004) [56] and anxiety disorder (Sagaspe et al., 2006) [56] symptoms in a sample of healthy adolescents following sleep deprivation. Excessive sleep duration, such as narcolepsy, has also been found to be associated with an increased occurrence of depression and anxiety in adolescents. For example, Kaplan and Harvey (2009) reported that narcolepsy may be an important mechanism contributing to the maintenance of symptoms of mood disorders [56]. These findings suggest that the prevention and optimisation of adolescent anxiety and depressive symptoms requires attention on the impact of adolescent sleep duration. However, this study did not use an objective instrument to measure sleep and therefore could not determine the duration of continuous sleep in adolescents, which also needs to be further explored in depth.

This study found that the greater the number of 24-HMG recommendations met, the lower the risk of depression and anxiety in adolescents. Meeting PA + sleep or meeting ST + sleep of the 24-HMG recommendations reduced the risk of anxiety and depression in adolescents compared to not meeting any of the recommendations. This is consistent with previous findings from Zhu et al. (2019) [48]. The results of this study suggest that the combined form of meeting PA + sleep or meeting ST + sleep may have a significant role in the prevention of anxiety and depression. Meeting PA+ST + sleep was more effective in reducing the risk of depression and anxiety in adolescents.

Meeting the 24-HMG recommendation categories differed in their effect on reducing the risk of depression or anxiety in adolescents. For example, meeting one 24-HMG recommendation for sleep duration alone significantly reduced the risk of depression and anxiety in adolescents. In two 24-HMG recommendations, meeting only physical activity time and sleep time together, and meeting both screen time and sleep time together significantly reduced the risk of adolescents developing anxiety. This further analysis, in which specific behavioural combinations are most beneficial to adolescents’ mental health, enables the implementation of more effective interventions. This suggests that the promotion and guidance of these specific behavioural combinations should be taken into account when implementing integrated or holistic interventions with adolescents, and that priority should be given to guiding which behavioural recommendations to meet if 24-HMG cannot be met in its entirety.

To our knowledge, this is the first time that the effects of meeting specific behavioural combination categories of the 24-HMG recommendations on adolescent mental health have been observed in a sample of Chinese adolescents, and is a further development and extension of the findings of Lu et al.’s (2021) study of children [27]. Our study could add new research evidence to the field of research on promoting adolescent mental health. This study measured the association between meeting physical activity, screen time and sleep time recommendations in the 24-HMG and the risk of developing anxiety and depression in Chinese adolescents in grade K8, using data from the national adolescent sample of the CEPS and using the Canadian 24-HMG as a reference standard. From the results of this study, it was suggested that encouraging an optimal combination of healthy behaviours may be an effective intervention to reduce anxiety and depression in adolescents. However, there were limitations to this study. Firstly, 55% of the schools surveyed in CEPS were in the eastern region of China, 19% in the central region and 24% in the western region. Due to the limitations of this sampling method of the CEPS, the results of this study need to be further validated with data from other large samples of Chinese adolescents. Secondly, the data in this study were derived from self-reported data of adolescents, and therefore the findings may be subject to bias from meetings and social expectations. Thirdly, the instruments used in this study to measure anxiety and depression were not derived from commonly used psychometric instruments of anxiety or depression. Although the validity of the CEPS’s approach to measuring adolescent mental health has been validated in a number of studies, we were unable to obtain additional data evidence from its development process. Fourthly, the measurements of physical activity, sedentary behaviour and sleep in this study were not based on objective measurement work. This may have implications for the stability of the estimates from this study. Fifthly, this study selected few control covariates and only examined adolescents’ sex, age, ethnicity, single child, residence, father’s highest level of education, mother’s highest level of education, perceived household economic status, and BMI were controlled for covariates. Previous studies have found that academic stress and social support may all have an impact on adolescent depression or anxiety, and the study did not control for these covariates. Finally, due to the cross-sectional design used in this study, no conclusions could be drawn regarding causality. In future studies, we recommend the use of objective measures of physical activity, quantity and quality of sleep, and validate depression and anxiety questionnaires, such as the depression anxiety spectrum disorder questionnaire (DASS).

## 5. Conclusions

This study determined the risk of developing depression and anxiety in adolescents who met the recommendations for 24-HMG alone and in combination. Overall, meeting more of the recommendations in the 24-HMG was associated with lower anxiety and depression risk outcomes in adolescents. For boys, reducing the risk of depression and anxiety can be prioritised by meeting PA + ST + sleep, meeting ST+ sleep and meeting sleep in the 24-HMG. For girls, reducing the risk of depression and anxiety may be preferred by meeting PA + ST + sleep, meeting PA + sleep and meeting sleep in 24-HMG.However, only a small proportion of adolescents met all recommendations, highlighting the need to promote and support adherence to these behaviours.

## Figures and Tables

**Figure 1 ijerph-20-03167-f001:**
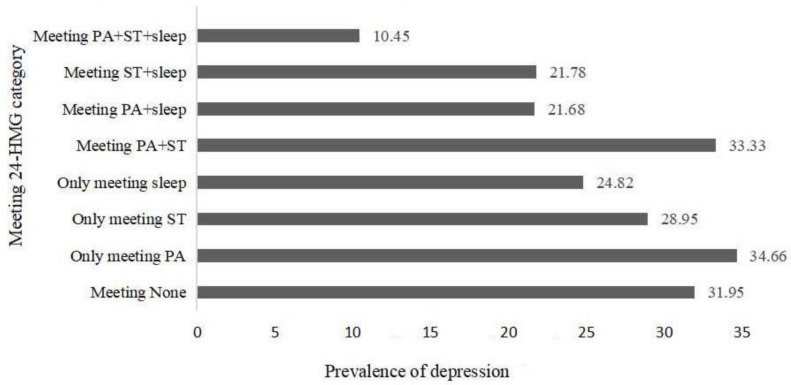
Prevalence of anxiety in different groups of adolescents.

**Figure 2 ijerph-20-03167-f002:**
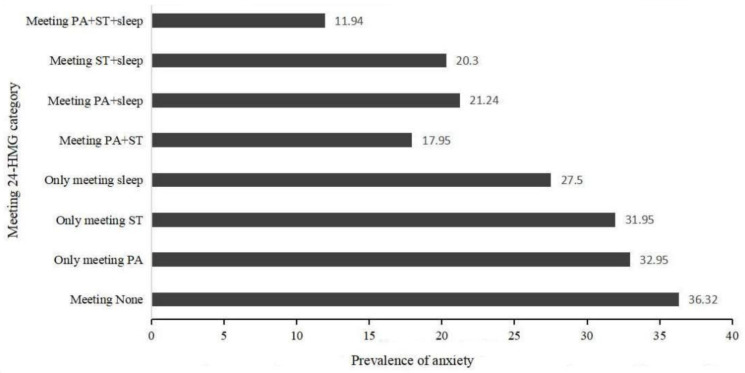
Prevalence of depression in different groups of adolescents.

**Table 1 ijerph-20-03167-t001:** Contents of 24-HMG.

Physical Activity Guidelines	Sleep Guidelines	Sedentary Guidelines Recommendations
An accumulation of at least 60 min per day of moderate to vigorous physical activity involving a variety of aerobic activities.	Uninterrupted 9 to 11 h of sleep per night for those aged 5–13 years and 8 to 10 h per night for those aged 14–17 years	No more than 2 h per day of recreational screen time
Vigorous physical activities and muscle and bone strengthening activities should each be incorporated at least 3 days per week	Consistent bed and wake-up times	Limited sitting for extended periods
Several hours of a variety of structured and unstructured light physical activities		

**Table 2 ijerph-20-03167-t002:** Characteristics of Chinese adolescents.

	Mean ± SD/%			Gender Differences	
	Total (*n* = 9420)	Boys (*n* = 5163)	Girls (*n* = 4257)	t/X^2^	*p*
Age(y)	14.53 ± 0.69	14.60 ± 0.67	14.45 ± 0.71	9.821	**<0.001**
Ethnicity				106.215	**<0.001**
Han	87.08	83.85	91.00		
Non-Han	12.92	16.15	9.00		
Single child				19.433	**<0.001**
Yes	44.86	47.06	42.42		
No	55.14	52.94	57.58		
Residence				4.134	**0.042**
City	47.78	46.76	48.91		
Rural	52.22	53.24	51.09		
Father’s highest education				34.382	**<0.001**
≤Junior middle school	58.52	61.15	55.34		
Senior middle school or vocational school	25.71	24.50	27.18		
≥College	15.76	14.35	17.48		
Mother’s highest education				18.960	**<0.001**
≤Junior middle school	64.66	64.49	62.44		
Senior middle school or vocational school	22.11	21.36	23.02		
≥College	13.23	12.14	14.54		
Perceived household economic status				16.672	**<0.001**
Lower income	20.69	22.03	19.19		
Middle class	73.45	71.64	75.45		
Wealthy	5.87	6.33	5.36		
BMI				48.771	**<0.001**
Below overweight	87.28	85.11	89.92		
Above overweight	12.72	14.89	10.08		
Meeting PA				130.358	**<0.001**
No	94.61	92.19	97.53		
Yes	5.31	7.81	2.47		
Meeting ST				63.635	**<0.001**
No	79.68	82.68	76.04		
Yes	20.32	17.32	23.96		
Meeting Sleep				53.230	**<0.001**
No	39.46	39.12	43.50		
Yes	60.54	63.88	56.50		
Meeting the 24-HMG category				254.361	**<0.001**
Meeting None	28.70	26.28	31.64		
Only meeting PA	1.87	2.61	0.96		
Only meeting ST	8.47	6.82	10.48		
Only meeting sleep	46.71	50.03	42.68		
Meeting PA + ST	0.41	0.41	0.42		
Meeting PA + sleep	2.40	3.76	0.75		
Meeting ST + sleep	10.72	9.06	12.73		
Meeting PA + ST + sleep	0.71	1.03	0.33		
Depression				0.122	0.726
No	73.07	72.92	73.24		
Yes	26.93	27.08	26.76		
Anxiety				0.897	0.344
No	70.56	70.97	70.07		
Yes	29.44	29.03	29.93		

Sleep: sleep guidelines; ST: screen time guidelines; PA: physical activity guidelines; *p*-values in bold indicate significant levels < 0.05.

**Table 3 ijerph-20-03167-t003:** Meeting one recommendation of the 24-HMG in relation to anxiety and depression in the sample.

	Total		Boys		Girls	
	OR (95%CI)	*p*	OR (95%CI)	*p*	OR (95%CI)	*p*
Depression						
Only meeting PA	0.97 (0.69~1.37)	0.890	0.97 (0.66~1.45)	0.918	0.85 (0.42~1.71)	0.663
Only meeting ST	0.85 (0.71~1.02)	0.073	0.83 (0.63~1.09)	0.186	0.87 (0.69~1.10)	0.259
Only meeting sleep	0.65 (0.58~0.73)	<0.001	0.60 (0.51~0.70)	<0.001	0.70 (0.60~0.82)	<0.001
Anxiety						
Only meeting PA	1.14 (0.82~1.60)	0.418	1.17 (0.79~1.73)	0.409	0.96 (0.48~1.92)	0.395
Only meeting ST	0.86 (0.72~1.03)	0.124	0.80 (0.61~1.06)	0.137	0.91 (0.72~1.16)	0.429
Only meeting sleep	0.68 (0.61~0.76)	<0.001	0.64 (0.54~0.74)	<0.001	0.73 (0.62~0.86)	<0.001

Sleep: sleep guidelines; ST: screen time guidelines; PA: physical activity guidelines; OR: odds ratio; CI: confidence interval. Ref: reference group; total models controlled for sex, age, ethnicity, single child, residence, father’s highest level of education, mother’s highest level of education, perceived household economic status, BMI. Boys and girls models controlled for age, ethnicity, single child, residence, father’s highest level of education, mother’s highest level of education, perceived household economic status, BMI. Ref: meeting none.

**Table 4 ijerph-20-03167-t004:** Meeting two recommendations of the 24-HMG in relation to anxiety and depression in the sample.

	Total		Boys		Girls	
	OR (95%CI)	*p*	OR (95%CI)	*p*	OR (95%CI)	*p*
Depression						
Meeting PA+ST	0.45 (0.19~1.03)	0.060	0.68 (0.24~1.92)	0.472	0.23 (0.05~1.04)	0.058
Meeting PA + sleep	0.52 (0.37~0.74)	<0.001	0.55 (0.38~0.79)	0.002	0.29 (0.10~0.83)	0.022
Meeting ST + sleep	0.46 (0.35~0.51)	<0.001	0.42 (0.32~0.56)	<0.001	0.42 (0.33~0.54)	<0.001
Anxiety						
Meeting PA + ST	1.14 (0.57~2.26)	0.699	1.41 (0.56~3.54)	0.466	0.88 (0.31~2.49)	0.057
Meeting PA + sleep	0.59 (0.42~0.83)	0.003	0.64 (0.44~0.93)	0.019	0.24 (0.07~0.79)	0.021
Meeting ST + sleep	0.57 (0.48~0.48)	<0.001	0.54 (0.41~0.71)	<0.001	0.59 (0.47~0.76)	<0.001

**Table 5 ijerph-20-03167-t005:** Meeting three recommendations of the 24-HMG in relation to anxiety and depression in the sample.

	Total		Boys		Girls	
	OR (95%CI)	*p*	OR (95%CI)	*p*	OR (95%CI)	*p*
Depression						
Meeting PA + ST + sleep	0.27 (0.13~0.58)	0.001	0.20 (0.08~0.52)	0.001	0.52 (0.16~2.19)	0.006
Anxiety						
Meeting PA + ST + sleep	0.18 (0.07~0.45)	<0.001	0.08 (0.02~0.34)	0.001	0.60 (0.16~2.19)	0.006

## Data Availability

Data can be requested.

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
