# Peer review of "The Associations between Meeting 24-Hour Movement Guidelines (24-HMG) and Mental Health in Adolescents—Cross Sectional Evidence from China"

_ijerph, 2023, doi:10.3390/ijerph20043167_

Round 1
Reviewer 1 Report (Previous Reviewer 2)
The manuscript has been improved, but further revisions are required before publication in IJERPH. Most of the changes made (especially in the Results and Discussion sections) significantly improved the quality of the manuscript. The issues listed below need improvement:
Title: adolescent or adolescents?
Abstract: The abstract uses a structured style, but with headings, while it should be without headings - this has not been corrected. CEPS – abbreviation not explained in the abstract: instead of “from the China Education Tracking Survey 2014-2015” should be “from the China Education Tracking Survey (CEPS) 2014-2015” – line 18
Material and Metods: In subsections 2.2.1-2.2.3 questions are still quoted. Instead of quoting the questions, please state what they were about. The authors wrote: “Shapiro-Wilk tests were performed for measured variables, and variables satisfying a normal distribution were described using the mean ± SD.” Does this mean that all the data had a normal distribution?
Results: If there were differences between groups, the description of the results should state what they consisted of (lines 260-263). In which group of girls or boys a higher percentage of subjects realized the fulfillment of 24-HMG number and category.
Other comments include:
1) In many places throughout the work (in the abstract, keywords, main text and in the footnotes under the tables and in References), especially between sentences, after commas and colons, spaces are missing. Selected excerpts from the text are quoted below:
Line 17: „and depression.Methods:”
Line 26: „in adolescents.Results: Of”
Line 31: “in adolescents.Meeting sleep”
Line 44: “in combination.Overall, meeting”
Line 49: “in 24-HMG.However,”
Lines 51-52: “Keywords:24-Hour Movement Guidelines;physical activity time;screen time;depression; adoles-51 cents; anxiety”
Line 60: “the study[6]”
Line 82: “5-17 years [13].Previous”
Line 94: “14-17 years [212].Ojio et al.”
Line 160: “database of CEPS.The ethical”
Line 171: “in Table 1.This study”
Line 173: “recommendations[29].And”
Line 180: “exercise(PE)”
Line 210: “was 0.815.Six”
Line 211: “0.918.Raw scores”
Line 223: “education,body”
Line 230: “[335].BMI is” and “not overweight(including low”
Line 231: “overweight(including overweight”
Line 235: “variables. Shapiro-Wilk tests”
Line 236: “mean ± SD.Gender”
Line 245: “sleep,meeting”
Line 262: “status ,BMI”, “meeting PA,meeting ST,meeting sleep and”
Line 263: “category.There”
Line 269: „;p-values”
Line 264: “and depression.A description”
Lines 297-301: “reference group;Total”, ”sex,age,ethnicity,single child,residence,father’s” “education,mother’s”, “education,perceived”, “status,BMI.Boys”, “age,ethnicity,single”, “child,residence,father’s”, “education,mother’s”, “highest education,perceived”, “status,BMI.Ref:meeting”
Line 371: “depression.In this”
Line 373: “studies.A”
Line 375: “was 3% [346].The study of”
There are more such places in the discussion, conclusions and literature list (e.g. lines: 377, 386, 387, 389, 390, 393, 399, 403, 409, 411, …., 525, 559, 562, …..)
2) Line 149: “method. the CEPS” - instead of "the" it should be "The"
3) Line 252: instead of "9420" it should be "9,420"
4) In table 2: instead of
Total (n=9420) |
Boys (n=5163) |
Girls (n=4257) |
should be
Total (n=9,420) |
Boys (n=5,163) |
Girls (n=4,257) |
5) Line 373: instead of "k8" it should be "K8"
6) In several places, activity has changed to activity? I don't understand this change. Was it supposed to be excercise?
7) PA or PE? - this should be standardized. From line 83 the abbreviation was changed - for what purpose?
8) Line: 164: http://ceps.ruc.edu.cn/ - missing from references.
Author Response
Dear expert,
Thank you very much for confirming the results of our revision and for once again giving us such detailed and sensible advice.
We respond to your suggestions as follows.
1.Title: adolescent or adolescents?
We have amended the title to adolescents.
2.Abstract: The abstract uses a structured style, but with headings, while it should be without headings - this has not been corrected. CEPS – abbreviation not explained in the abstract: instead of “from the China Education Tracking Survey 2014-2015” should be “from the China Education Tracking Survey (CEPS) 2014-2015” – line 18
We didn't quite understand what you meant by structured style, but we have re-referenced the empty template given by the magazine and made changes. Could you please help us to see if we have understood correctly? Please refer to the abstract section.
"from the China Education Tracking Survey 2014-2015" has been amended to "from the China Education Tracking Survey (CEPS) 2014-2015"
3.Material and Metods: In subsections 2.2.1-2.2.3 questions are still quoted. Instead of quoting the questions, please state what they were about. The authors wrote: “Shapiro-Wilk tests were performed for measured variables, and variables satisfying a normal distribution were described using the mean ± SD.” Does this mean that all the data had a normal distribution?
The descriptions in 2.2.1-2.2.3 have been revised and you are invited to review 2.2.1-2.2.3.The "Shapiro-Wilk test was performed on the measured variables and variables satisfying a normal distribution were described by mean ± SD" in the statistical analysis has been revised.Please review lines 191-192.
4.Results: If there were differences between groups, the description of the results should state what they consisted of (lines 260-263). In which group of girls or boys a higher percentage of subjects realized the fulfillment of 24-HMG number and category.
We have deleted the original lines 260-263 and changed the description. Gender actually has no effect on ORS for anxiety and depression, and we may have been inaccurate in describing it that way.Please review the revised sections in 3.4, 3.5 and 3.6.
Boys and girls with meeting 24-HMG category differences are described in our supplement in 3.1 and 3.2.Please review.
5.Line 373: instead of "k8" it should be "K8"
Already revised.
6.In several places, activity has changed to activity? I don't understand this change. Was it supposed to be excercise?
"exercise"" has been changed to "activity"
7.PA or PE? - this should be standardized. From line 83 the abbreviation was changed - for what purpose?
PE has been modified to PA
8.Line: 164: http://ceps.ruc.edu.cn/ - missing from references.
http://ceps.ruc.edu.cn/ is a link to the data site, so we have not placed a reference.
In addition, we have read through the whole text to correct punctuation errors, problems with spaces, and the errors you pointed out in the text. Thank you very much for being so meticulous and we are ashamed of our carelessness.
All our revisions have been marked in red using the revision model. Thank you again for your selfless help, we have benefited greatly from these suggestions.
Reviewer 2 Report (Previous Reviewer 3)
The authors have made substantial changes that improve the manuscript. I do not have further comments.
Author Response
Dear expert,
Thank you very much for confirming the results of our revision.
This manuscript is a resubmission of an earlier submission. The following is a list of the peer review reports and author responses from that submission.
Round 1
Reviewer 1 Report
Thank you for an interesting study.
The manuscript is a little crowded and it would be worthwhile to re evaluate the set up so the results read more easily and is not as confusing:
Perhaps report each intervention separately then 2 of3 interventions then 3 of 3 interventions.
Did you draw any major conclusions? Exercise > Sleep > ST for wellbeing?
Author Response
Dear experts,
Thank you very much for your recognition of our research results and for your help.
First of all, I am very sorry that the reworking of our thesis has come so late due to our work and health reasons.
Here is our response to your suggestion:
Your suggestion:
Perhaps report each intervention separately then 2 of 3 interventions then 3 of 3 interventions.
Our response:
With reference to your suggestions, we have rearranged the data processing. It has been divided into one recommendation, two recommendations and three recommendations. Please review 3.3 - 3.5.
Your suggestion:
Did you draw any major conclusions? Exercise > Sleep > ST for wellbeing?
Our response:
We have re-described and compared the effects of the different recommended combinations, please refer you to the data and data descriptions in sections 3.3 - 3.5.
As a result of these revisions, we also revised our abstract and conclusions. Also as other experts have suggested a number of changes, we have integrated these suggestions for the revision of the paper. We have used the red revision mode for all revisions to facilitate your review. However, we are sorry for any new workload this may have caused you.
Finally, thank you again for your help and guidance, we have benefited greatly from it.
Reviewer 2 Report
Overall, the manuscript presented for review focuses on an important issue. Anxiety and depression affect all age groups, including adolescents, and cause somatic complications and problems in daily functioning, and have a significant impact on quality of life. The study of the determinants of mental health, especially in adolescents, with the association of meeting the 24-hour movement guidelines, is very valuable and needed. The research undertaken is therefore a response to the demand in this area.
The layout of the article is typical of original papers and includes all required parts. Some sub-sections are well-designed, but most sections need improvement - for some sections major revision.
In the last paragraph of the “Introduction”, the authors stated that the purpose of the study would be to determine whether the results obtained are related to gender. Admittedly, such results are presented in the table, but neither the discussion nor the conclusions address this aspect. The description of the results separately describes the results for girls and boys, without comparing them. Authors are requested to obligatorily complete this information in the main text and the abstract.
In the "Results" section, the authors have limited themselves to merely repeating the values given in the tables without any description in the form of a comparison. Table 1 is very extensive and contains results that were not cited in their description. In addition, this table contains repetitions of the results for "Meeting 24-HMG category” and “Meeting number of 24-HMG”.
The description of the results lacked commentary on the results shown in the figure. The authors limited themselves to the information that: “A forest plot of relationship between meeting 2HMG category and” depression or anxiety “was shown in Figure 1” (a or b).
All results are presented as means and standard deviations. Did all the data have a normal distribution? What test was used to check this? The methodology lacks information on this subject.
“Abstract” section lacks information on instrument used to study anxiety and depression. In line 35, it is worth adding what recommendations. In line 37, instead of "these," state what specific behaviors.
In the Material and methods section, please complete the ethics committee approval number. It is worth stating whether the study participants were informed about the purpose and content of the study and who supervised the study in the schools.
The characterization of the measures lacks justification for why the Canadian 24-HMG was used, and instead of quoting the questions, please state what they were about (subsections 2.2.1-2.2.3). Subsection 2.2.4 lacks information on what the questions used to assess depression and anxiety were about - only the numbers of the questions used are given. What does "overweight or more" mean? Overweight or obese? And there were no underweight participants?
The first paragraph of the “Discussion” should deal with a brief presentation of all results obtained in the study - please complete this. The same information is repeated several times in the discussion. Please eliminate it. The discussion needs to be supplemented with information on the factors studied, especially the association with gender.
The conclusions do not fully reflect the purpose of the study and require mandatory correction.
In the list of references, the full names of the journals are given everywhere (in many places the members of the names are lowercase), not abbreviations. This should be corrected. In the manuscript, I found no reference to the 16 literature items.
Other comments include:
1) In many places throughout the work (in the abstract, keywords, main text and in the footnotes under the tables), especially between sentences, after commas and colons, spaces are missing.
2) All abbreviations used in the manuscript should be explained at the first place of their use. In later parts of the manuscript, these explanations should no longer be repeated.
3) The abstract uses a structured style, but with headings, while it should be without headings.
4) Line 50: “movementvity” or movement?
5) Line 89: “And” at the beginning of a sentence?
6) Line 101: instead of a dot there should be a semicolon or comma.
7) Line 112: instead of "the" it should be "The".
8) Line 121: instead of "9420" it should be "9,420".
9) Lines 239 and 255: instead of "Meeting" it should be "meeting".
10) Line 280: % at the beginning of a sentence?
11) Line 288: instead of "liu" it should be "Liu".
12) Line 289: instead of "k9-k12" it should be "K9-K12".
13) Line 292: instead of "3772" it should be "3,772".
14) Lines 381-388: in the conclusions should be full names, not abbreviations of the tool and the factors studied.
15) Line 388: Remove the last sentence from the conclusions. This is a hint from the Template.
16) In the supplementary file and in the unpublished material is the same table. No table number and title above the table.
Author Response
Dear experts,
Thank you very much for your recognition of our research results and for your help.
First of all, I am very sorry that the reworking of our thesis has come so late due to our work and health reasons.
Here is our response to your suggestion:
Your suggestion:
In the last paragraph of the “Introduction”, the authors stated that the purpose of the study would be to determine whether the results obtained are related to gender. Admittedly, such results are presented in the table, but neither the discussion nor the conclusions address this aspect. The description of the results separately describes the results for girls and boys, without comparing them. Authors are requested to obligatorily complete this information in the main text and the abstract.
Our response:
We have revised the description of the data in the abstract and body in response to your suggestions. Please review the summary and the description of the data in sections 3.3-3.5. As another expert has suggested a strategy for presenting the data, we have adjusted the framework for presenting the data into one recommendation, two recommendations and three recommendations.
Your suggestion:
In the "Results" section, the authors have limited themselves to merely repeating the values given in the tables without any description in the form of a comparison. Table 1 is very extensive and contains results that were not cited in their description. In addition, this table contains repetitions of the results for "Meeting 24-HMG category” and “Meeting number of 24-HMG”.
Our response:
In response to your suggestion, we have supplemented the description in Table 1 and removed the data in the Meeting number of 24-HMG section.
Your suggestion:
The description of the results lacked commentary on the results shown in the figure. The authors limited themselves to the information that: “A forest plot of relationship between meeting 2HMG category and” depression or anxiety “was shown in Figure 1” (a or b).
Our response:
We have removed this part of the forest diagram to allow the results to be more focused on the categories of recommendations.
Your suggestion:
All results are presented as means and standard deviations. Did all the data have a normal distribution? What test was used to check this? The methodology lacks information on this subject.
Our response:
In the statistics section, we have added our strategy and methodology for describing the data. Please review 2.3.
Your suggestion:
“Abstract” section lacks information on instrument used to study anxiety and depression. In line 35, it is worth adding what recommendations. In line 37, instead of "these," state what specific behaviors.
Our response:
Sources of data on anxiety and depression that we have added to the abstract. We have corrected the error in lines 35 and 37 of your original statement.
Your suggestion:
In the Material and methods section, please complete the ethics committee approval number. It is worth stating whether the study participants were informed about the purpose and content of the study and who supervised the study in the schools.
Our response:
We use secondary data, ethics and more detailed information about the project can be found on the official website of the data. All subjects and their parents signed an informed consent form. This is described in our lines 136-142.
Your suggestion:
The characterization of the measures lacks justification for why the Canadian 24-HMG was used, and instead of quoting the questions, please state what they were about (subsections 2.2.1-2.2.3). Subsection 2.2.4 lacks information on what the questions used to assess depression and anxiety were about - only the numbers of the questions used are given. What does "overweight or more" mean? Overweight or obese? And there were no underweight participants?
Our response:
We added a full description of the Canadian 24-HMG and explained what we had chosen. A more detailed description of the overweight indicator is then provided. We invite you to review section 2.2.
Your suggestion:
The first paragraph of the “Discussion” should deal with a brief presentation of all results obtained in the study - please complete this. The same information is repeated several times in the discussion. Please eliminate it. The discussion needs to be supplemented with information on the factors studied, especially the association with gender.
Our response:
We have removed the first paragraph of the discussion and then added a discussion of gender, please review the discussion section.
Your suggestion:
The conclusions do not fully reflect the purpose of the study and require mandatory correction.
Our response:
We have reworked the Conclusion section and ask you to review it.
Your suggestion:
In the list of references, the full names of the journals are given everywhere (in many places the members of the names are lowercase), not abbreviations. This should be corrected. In the manuscript, I found no reference to the 16 literature items.
Our response:
We have reformatted the references.
Your suggestion:Other comments include:
1) In many places throughout the work (in the abstract, keywords, main text and in the footnotes under the tables), especially between sentences, after commas and colons, spaces are missing.
2) All abbreviations used in the manuscript should be explained at the first place of their use. In later parts of the manuscript, these explanations should no longer be repeated.
3) The abstract uses a structured style, but with headings, while it should be without headings.
4) Line 50: “movementvity” or movement?
5) Line 89: “And” at the beginning of a sentence?
6) Line 101: instead of a dot there should be a semicolon or comma.
7) Line 112: instead of "the" it should be "The".
8) Line 121: instead of "9420" it should be "9,420".
9) Lines 239 and 255: instead of "Meeting" it should be "meeting".
10) Line 280: % at the beginning of a sentence?
11) Line 288: instead of "liu" it should be "Liu".
12) Line 289: instead of "k9-k12" it should be "K9-K12".
13) Line 292: instead of "3772" it should be "3,772".
14) Lines 381-388: in the conclusions should be full names, not abbreviations of the tool and the factors studied.
15) Line 388: Remove the last sentence from the conclusions. This is a hint from the Template.
16) In the supplementary file and in the unpublished material is the same table. No table number and title above the table.
Our response:
We have made changes with your suggestions in mind and thank you very much for your so detailed advice.
Also as other experts have suggested a number of changes, we have integrated these suggestions for the revision of the paper. We have used the red revision mode for all revisions to facilitate your review. However, we are sorry for any new workload this may have caused you.Finally, thank you again for your help and guidance, we have benefited greatly from it.
Reviewer 3 Report
INTRODUCTION
I would advise a shorter introduction. Explain the 24-HMG briefly. Mention prevalence of anxiety and depression among adolescents. Explain the risks associated with suffering from both mental conditions. Explain studies that have associated 24-HMG with both conditions and then justify the need for a study like this among adolescents Chinese.
METHODS
The authors used information derived from several “ad hoc” questions? Is there any information on the reliability and validity of the questions used for assessing the main study outcomes?
24-HMG recommends the performance of 60 min/day of moderate to vigorous PA. Judging from the used questions, the authors can not identify whether this goal is achieved. Similarly, 24-HMG asks for uninterrupted sleep. Please, elaborate on this.
RESULTS
I am confused. Students are at grade K8, but age range is 13-18? How can this be possible? Did K8 involve several educational levels?
Would it be possible to show the percentage of students with depression and with anxiety that meet/not meet 24-HMG recommendations item by item (PA, sleep..) and as a whole (number or items met/not met).
DISCUSSION
This section is well written. I miss information on the importance of having siblings. The authors did register this data, and some studies have relates having siblings with motor performance and PA.
The mean weak points are well-addressed in the limitations sections.
Line 317 approach is repeated.
CONCLUSION
Meeting one recommendation (sleep). Does this mean that meeting one recommendation (PA or ST) does not have the same effect? If so, this should be highlighted.
Author Response
Dear experts,
Thank you very much for your recognition of our research results and for your help.
First of all, I am very sorry that the reworking of our thesis has come so late due to our work and health reasons.
Here is our response to your suggestion:
Your suggestion:
INTRODUCTION
I would advise a shorter introduction. Explain the 24-HMG briefly. Mention prevalence of anxiety and depression among adolescents. Explain the risks associated with suffering from both mental conditions. Explain studies that have associated 24-HMG with both conditions and then justify the need for a study like this among adolescents Chinese.
Our response:
We have revised the Introduction section in response to your suggestions. We have added information about 24-HMG and the incidence of depression and anxiety, etc. Please review the changes marked in red in the Introduction section.
Your suggestion:
METHODS
The authors used information derived from several “ad hoc” questions? Is there any information on the reliability and validity of the questions used for assessing the main study outcomes?
- HMG recommends the performance of 60 min/day of moderate to vigorous PA. Judging from the used questions, the authors can not identify whether this goal is achieved. Similarly, 24-HMG asks for uninterrupted sleep. Please, elaborate on this.
Our response:
The measures of physical activity, sedentary behaviour and sleep in this study were not based on objective measurement exercises. This may have an impact on the stability of the estimates in this study. However, because we are using secondary data, the measures are subject to the limitations of the original data design. However, the reliability and validity of these questions have been tested by the CEPS staff, and attention to these data has led to a number of high level academic publications. However, we have also included it as a research deficiency in the last paragraph of the discussion.Further information can be found on the CEPS database website. http://ceps.ruc.edu.cn/.
Your suggestion:
RESULTS
I am confused. Students are at grade K8, but age range is 13-18? How can this be possible? Did K8 involve several educational levels?
Would it be possible to show the percentage of students with depression and with anxiety that meet/not meet 24-HMG recommendations item by item (PA, sleep..) and as a whole (number or items met/not met).
Our response:
CEPS (2014-2015) records data for students in grade 8, which is indeed 13-18 years of age. We re-examined the data and found that there were 5 students aged 18 and 86 students aged 17. This group of students is mainly from rural areas. In rural areas of China, due to the economic conditions of families, some students do start school later and even repeat grades, which may also account for the older age of the students. However, we have not artificially removed this part of the data and we believe that they are also reasonable. This is because the student's date of birth is extracted from the school's school card. However, we have removed this 13-18 description method and the values show the average age.
With reference to your suggestions, we have added the prevalence of depression and anxiety for the different categories recommended. We kindly ask you to review section 3.2.
Your suggestion:
DISCUSSION
This section is well written. I miss information on the importance of having siblings. The authors did register this data, and some studies have relates having siblings with motor performance and PA.
The mean weak points are well-addressed in the limitations sections.
Line 317 approach is repeated.
Our response:
Our aim was primarily to highlight the relationship between different categories of recommendation and the risk of developing depression and anxiety. In our hidden data, there was no significant difference in the effect of the status of single child on the risk of developing depression and anxiety in our analytical model.
Your suggestion:
CONCLUSION
Meeting one recommendation (sleep). Does this mean that meeting one recommendation (PA or ST) does not have the same effect? If so, this should be highlighted.
Our response:
We have restructured the description of the conclusions and invite you to review the conclusions section.
Also as other experts have suggested a number of changes, we have integrated these suggestions for the revision of the paper. We have used the red revision mode for all revisions to facilitate your review. However, we are sorry for any new workload this may have caused you.Finally, thank you again for your help and guidance, we have benefited greatly from it.
Reviewer 4 Report
This study adds to existing literature by examining the relationship between meeting the 24-HMG and mental health in adolescents aged between 13-18 years.
A nice aspect of the manuscript is the detailed description of existing data/literature in the Introduction.
Lines 328-353: Suggest the authors re-write this section of the Discussion. A strong discussion is required to emphasise the importance of meeting the sleep guidelines, given that sleep disturbances have a strong link to mental disorders.
Please see specific comments within the manuscript.

Author Response
Dear experts,
Thank you very much for your recognition of our research results and for your help.
First of all, I am very sorry that the reworking of our thesis has come so late due to our work and health reasons.
Here is our response to your suggestion:
Your suggestion:
Lines 328-353: Suggest the authors re-write this section of the Discussion. A strong discussion is required to emphasise the importance of meeting the sleep guidelines, given that sleep disturbances have a strong link to mental disorders.
Our response:
We have made changes in the light of your suggestions and ask you to review lines 392-412.
We have made amendments to your suggestions in the pdf, but because the points are so detailed we can't respond better on a line-by-line basis. Hopefully you will see it when it is re-examined.
Also as other experts have suggested a number of changes, we have integrated these suggestions for the revision of the paper. We have used the red revision mode for all revisions to facilitate your review. However, we are sorry for any new workload this may have caused you.Finally, thank you again for your help and guidance, we have benefited greatly from it.